# Electrical Optimization Method Based on a Novel Arrangement of the Magnetic Navigation System with Gradient and Uniform Saddle Coils

**DOI:** 10.3390/s22155603

**Published:** 2022-07-27

**Authors:** Sungjun Kim, Mingyu Cho, Seyeong Im, Joongho Yun, Jaekwang Nam

**Affiliations:** Department of Robotics, Kwangwoon University, Seoul 01897, Korea; kxt1234@naver.com (S.K.); alsrb77022@naver.com (M.C.); syl1219ss@naver.com (S.I.); tygh7892@naver.com (J.Y.)

**Keywords:** magnetic navigation system, saddle coil, optimization, magnetic robot, magnetic field

## Abstract

The magnetic navigation system (MNS) with gradient and uniform saddle coils is an effective system for manipulating various medical magnetic robots because of its compact structure and the uniformity of its magnetic field and field gradient. Since each coil of the MNS was geometrically optimized to generate strong uniform magnetic field or field gradient, it is considered that no special optimization is required for the MNS. However, its electrical characteristics can be still optimized to utilize the maximum power of a power supply unit with improved operating time and a stronger time-varying magnetic field. Furthermore, the conventional arrangement of the coils limits the maximum three-dimensional (3D) rotating magnetic field. In this paper, we propose an electrical optimization method based on a novel arrangement of the MNS. We introduce the objective functions, constraints, and design variables of the MNS considering electrical characteristics such as resistance, current density, and inductance. Then, we design an MNS using an optimization algorithm and compare it with the conventional MNS; the proposed MNS generates a magnetic field or field gradient 22% stronger on average than that of the conventional MNS with a sevenfold longer operating time limit, and the maximum three-dimensional rotating magnetic field is improved by 42%. We also demonstrate that the unclogging performance of the helical robot improves by 54% with the constructed MNS.

## 1. Introduction

Magnetic robots and tools have attracted considerable attention as alternatives to conventional medical devices such as catheters, endoscopes, grippers, and forceps [1,2,3,4,5,6,7,8]. Remote magnetic actuation not only provides an unlimited operation time, but also provides the advantage of being able to miniaturize the robot because it does not require batteries. As a part of this miniaturization, magnetic particles also have been studied for various medical purpose such as targeted drug delivery, embolization, and imaging technology [9,10,11,12,13]. These advantages render magnetic robots suitable for minimally invasive medical devices that must operate for a long time inside small and narrow human organs. However, magnetic robots require a magnetic navigation system (MNS), which generates an external magnetic field to the magnetic robots. MNSs can be classified into two types, with and without a magnetic core. The MNSs with a core are particularly advantageous in generating a strong magnetic field to manipulate small magnetic nanoparticles or microrobots [14,15,16,17] because the core amplifies the generated magnetic field. However, the coreless MNSs allow a magnetic robot to be controlled without localization because they generate uniform magnetic field or field gradient over a large workspace. Magnetic resonance imaging (MRI) or magnetic particle imaging (MPI) devices can also be used for manipulation, but they are not satisfactory for various medical magnetic robots because their functions are focused on imaging [8,9,10].

A simple coreless MNS combines several pairs of Helmholtz coils (HCs) and Maxwell coils (MCs). These circular coils are structurally simple; hence, many researchers have used this combination to actuate magnetic robots [18,19,20,21,22,23,24,25]. Three pairs of HCs are the basic combination for generating a three-dimensional (3D) magnetic field [8,9,10]. Several pairs of MCs can also be integrated with the HCs to generate both a magnetic field and a field gradient [21,22,23]. However, the combination of multiple pairs of circular coils is geometrically inefficient. These combinations produce large empty spaces between pairs of coils. To address this issue, rotatable MNSs were developed [24,25]; instead of increasing the number of coils, the coils can be rotated to ensure the required degree of freedom (DOF). However, this method makes the MNS complex because it requires additional electromechanical systems. In addition, uniform saddle coils (USCs) and gradient saddle coils (GSCs) have been developed [26]. These coils can surround circular coils to form a compact cylindrical structure without requiring extra space. The cylindrical shape is appropriate for accommodating the human body and is used in other similar medical devices, such as magnetic resonance imaging (MRI) machines and computerized tomography (CT) scanners. There are various combinations of circular and saddle coils [26,27,28,29,30]. In particular, the MNS composed of five pairs of coils (two USCs, a GSC, an HC, and an MC) can generate a 3D rotation and two-dimensional (2D) translation of magnetic robots [27]. Since each coil of the MNS was geometrically optimized to generate strong uniform magnetic field or field gradient, it is considered that no special optimization is required for the MNS. However, even if coils have the same geometry, electronical characteristics, such as resistance, current density, and inductance, can be different depending on the number of turns and thickness of wires. These electronic characteristics affect the performance of the MNS. For example, if the resistance of a coil is too large or too small, a power supply unit cannot provide maximum output power owing to its current or voltage limit. If a coil has high current density, it reduces the operating time limit owing to the rapid temperature rise of the coil. If a coil has large inductance, we obtain a small time-varying magnetic field owing to large inductance effect. Furthermore, the arrangement of the coils can affect the performance of the MNS for a 3D rotating magnetic field, but this has not been considered.

In this study, an electrical optimization method is proposed on the basis of a novel arrangement of the MNS with five pairs of coils. First, two pipes for installing the five pairs of coils were determined so as to have the required inner space. Then, we propose a novel arrangement of the five pairs of coils that enables the MNS to generate similar three-axis magnetic fields per current because the available amplitude of a 3D rotating magnetic field is restricted by the coil that generates the smallest magnetic field. Next, the objective functions, design variables, and constraints were defined for optimization, with some assumptions to simplify the calculation. The constraints include the inductance effect and current density of the coils such that the MNS can generate a strong time-varying magnetic field under a given temperature limit. The MNS was optimized using an optimization algorithm with the given constraints. Finally, the optimized MNS was constructed and verified. Then, the performance was compared with the conventional MNS. We confirmed that the optimized MNS could generate a stronger magnetic field in both static and dynamic conditions. The improved magnetic field can be used to enhance the quasi-static and dynamic motion of magnetic robots. As one example, we demonstrated that the optimized MNS can effectively improve the unclogging performance of a helical robot.

## 2. Novel Arrangement of the MNS

### 2.1. Magnetic Field and Field Gradient Generated by Each Pair of Coils

The magnetic field and field gradient from the MNS produce a magnetic torque and force for a magnetic robot to generate rotation and translation motion [28]. The torque and force can be expressed as
(1)$$\vec{F}=\left(\vec{m}\cdot \nabla \right)\vec{B},$$
(2)$$\vec{T}=\left(\vec{m}\times \vec{B}\right),$$
respectively, where $\vec{m}$ is the magnetic moment of the robot and $\vec{B}$ is the external magnetic field. This external magnetic field can be generated by five pairs of coils, as shown in Figure 1, and the three components of the magnetic field vector can be expressed as follows [26]:(3)$${\vec{B}}_{MNS}=\left[\begin{array}{c} B_{h}+\left(G_{g}+G_{m}\right)x\\ B_{uy}+\left(-2.4398G_{g}-0.5G_{m}\right)y\\ B_{uz}+\left(1.4398G_{g}-0.5G_{m}\right)z \end{array}\right],$$
$$B_{h}={\left(4/5\right)}^{3/2}N_{h}I_{h}\mu_{0}/r_{h},\;B_{uy}=0.6004N_{uy}I_{uy}\mu_{0}/r_{uy},\;B_{uz}=0.6004N_{uz}I_{uz}\mu_{0}/r_{uz},\;G_{m}=\left(16/3\right){\left(3/7\right)}^{5/2}N_{m}I_{m}\mu_{0}/r_{m}^{2},\;G_{g}=0.3286N_{g}I_{g}\mu_{0}/r_{g}^{2},$$
where $N_{k}$, $I_{k}$, and $r_{k}$ are the number of turns, current, and radius of the *k*-th coil, respectively, the subscripts *h*, *m*, *g*, *uy*, and *uz* represent the HC, MC, GSC, and *y*- and *z*-directional USCs, respectively, and $\mu_{0}$ is the magnetic permeability of the free space. In Equation (3), *x*, *y*, and *z* are assumed to be negligible around the center of the MNS, where the workspace is located. Then, the magnetic torque is generated by three components ($B_{h}$, $B_{uy}$, and $B_{uz}$), whereas the magnetic force is generated by two components ($G_{g}$ and $G_{m}$).

### 2.2. Frame of the MNS

The frame of the MNS can be made of pipes owing to its cylindrical structure of the MNS. Figure 2a shows the configuration of the MNS using two pipes. Because the coils can be attached to both sides of the pipes, the two pipes are sufficient to install all coils. The diameter of each pipe was determined by considering the volume and magnetic field of the MNS. If the diameter is too small, the MNS cannot provide sufficient inner space for a magnetic robot. In contrast, if the diameter is too large, the coils cannot generate a sufficiently strong magnetic field to actuate the magnetic robot. According to this idea, the inner and outer diameters of the pipes ($d_{in}$ and $d_{out}$) were determined to be 31 and 46 cm, respectively. The thickness of the pipes ($t_{p}$) was 5 mm, and the gap between the pipes ($d_{gap}$) was 7 cm. The pipes had four square windows to observe the inner space of the MNS during the experiment.

### 2.3. Arrangement of the MNS

Various mechanisms of magnetic robots are based on a 3D rotating magnetic field; however, the available amplitude of a 3D rotating magnetic field is restricted by the coil that generates the smallest magnetic field. Thus, the three components of the magnetic field ($B_{h}$, $B_{uy}$, and $B_{uz}$) should have the same values. In Equation (3), the HC can generate a larger magnetic field than the USC under the same conditions ($N_{k}$, $I_{k}$, and $r_{k}$). Therefore, the HC should be placed outside the USC. Then, $B_{h}$ can be equal to $B_{uy}$ or $B_{uz}$ with a larger radius. However, because USCy and USCz are of the same type of coil, if the same $N_{k}$ and $I_{k}$ are given, the magnetic field generated by the outer coil is inevitably smaller. To solve this problem, we divided USCz and placed USCy between them; USCz1 and USCz2 were connected in series. Then, by adjusting the ratio of USCz1 to USCz2, USCy and USCz could generate the same magnetic field ($B_{uy}$ and $B_{uz}$).

Figure 2 shows this novel arrangement of the divided USCz. For two pipes, this arrangement was the only possible case. The MC and GSC were then placed in the empty spaces. The MC was placed on the outermost side of the MNS because it could be overlapped with the HC. In contrast, the GSC could be overlapped with USCy or USCz. The GSC could generate the strongest magnetic field gradient if it is placed with USCz; however, in this case, the GSC reduced the inner space of the MNS because the GSC was thicker than the divided coil (USCz2). Thus, we placed the GSC with USCy.

## 3. Electrical Optimization of the MNS

### 3.1. Objective Function and Design Variables of Each Coil

The MNS has three magnetic field components ($B_{h}$, $B_{uy}$, and $B_{uz}$) and two field gradient components ($G_{g}$ and $G_{m}$). We assume that the maximum output of the power supply unit ($P_{out}$) is utilized for each coil. Then, the five components of the MNS can be rewritten as
(4)$$B_{k,max}=\frac{n_{k}N_{k}\mu_{0}}{r_{k}}\sqrt{\frac{P_{out}}{R_{k}}}\;,$$
(5)$$G_{k,max}=\frac{n_{k}N_{k}\mu_{0}}{r_{k}^{2}}\sqrt{\frac{P_{out}}{R_{k}}}\;,$$
where $n_{k}$ and $R_{k}$ are the coefficient (in Equation (3)) and resistance of the *k*-th coil, respectively. These five components become the objective functions of the coils, and our goal is to maximize these values with several constraints, which are described in Section 3.2. In this study, we defined the design variables as the number of turns ($N_{k}$) and the thickness of the wire ($t_{k}$). We could then reorganize the objective functions using these design variables. In Equations (4) and (5), the resistance ($R_{k})$ of the *k*-th coil can be expressed as
(6)$$R_{k}=\frac{l_{k}}{A_{k}}\rho_{coil},$$
where $l_{k}$, $A_{k}$, and $\rho_{k}$ are the total length, cross-sectional area, and resistivity of the wound wire with $N_{k}$ turns, respectively. In Equation (6), the cross-sectional area ($A_{k})$ can be calculated assuming a circular wire, and the total length ($l_{k}$) can be calculated assuming that the wire is wound only at the center of the coil. Consequently, $l_{k}$ and $A_{k}$ can be calculated as follows:(7)$$l_{k\;}=c_{k}r_{k}N_{k},$$
$$c_{h\;}=c_{m}=4\pi ,\;c_{uy}=c_{uz}=22.3788,\;c_{g}=13.3704,$$
(8)$$A_{k}=\pi t_{k}^{2}/4,$$
where $c_{k}$ is the geometric coefficient of the *k*-th coil. The coefficients of $c_{h}$ and $c_{m}$ can be calculated considering the circular shape of the HC and MC, and the coefficients of $c_{uy}$, $c_{uz}$, and $c_{g}$ can be calculated considering the saddle shapes of the USC and GSC. Furthermore, $r_{k}$ can be calculated assuming that the bundle of wires forms a square cross-section, as shown in Figure 3. Although the actual cross-section is close to a circle, this assumption is valid because the radius of the bundle is very small compared to the radius of the coil. Then, $r_{k}$ can be expressed as
(9)$$r_{k}=0.5d_{k}\pm 0.5t_{k}\sqrt{N_{k}},$$
$$d_{h,m}=d_{out},\;d_{uz1}=d_{out}-2t_{p},\;d_{uy,\;g}=d_{in},\;d_{uz2}=d_{in}-2t_{p},$$
where $d_{k}$ is the outer or inner diameter of the pipe to which the *k*-th coil is attached. In Equation (9), the sign is determined on the basis of the location of the coil attached to the pipe. If the coil is attached to the outside, it has a positive sign (+); if the coil is attached to the inside, it has a negative sign (−). Finally, the objective function of the *k*-th coil can be rewritten using the two design variables ($N_{k}$ and $t_{k}$) by substituting Equations (6)–(9) into Equations (4) and (5).
(10)$$B_{k,max}=\frac{\mu_{0}n_{k}t_{k}}{{\left(0.5d_{k}\pm 0.5t_{k}\sqrt{N_{k}}\right)}^{3/2}}\sqrt{\frac{\pi N_{k}P_{out}}{4c_{k}\rho_{coil}}},$$
(11)$$G_{k,max}=\frac{\mu_{0}n_{k}t_{k}}{{\left(0.5d_{k}\pm 0.5t_{k}\sqrt{N_{k}}\right)}^{5/2}}\sqrt{\frac{\pi N_{k}P_{out}}{4c_{k}\rho_{coil}}}.$$

In Equations (10) and (11), the resistivity ($\rho_{k}$) can be changed depending on a temperature of the coil, but we assumed it as a constant value because we could not control the temperature without an additional cooling system.

### 3.2. Constraints of Each Coil

We considered several constraints to optimize the MNS. The first constraint was that the maximum values of the three components of the magnetic field are equal.
(12)$$B_{h,max}=B_{uy,max}=B_{uz,max}.$$

This constraint was described in Section 2.3. Using this constraint, the 3D rotating magnetic field can be maximized. We also considered the temperature rise of the coils because an insulated wire has an allowable temperature for safe use, which limits the available operating time of each coil. In addition, this temperature rise can deteriorate the performance of the MNS because of the increase in resistivity. Because the temperature of a coil is proportional to its current density, enlarging the cross-sectional area of each coil ($T_{k}^{2}$) can be the solution to increase the operating time limit. However, USCy, USCz1, and the GSC share a limited space between the pipes, as shown in Figure 2c. To enlarge the cross-sectional area without interference from the coils, we introduce the following constraints:(13)$$T_{uy}+T_{uz1}<d_{gap},$$
(14)$$T_{g}+T_{uz1}<d_{gap},$$
(15)$$0.5d_{gap}<T_{k}.$$

In Equation (15), $T_{uz1}$ and $T_{uz2}$ are not considered separately because they are connected in series. Instead, their summation ($T_{uz1}+T_{uz2}$) is considered as $T_{uz}$. We also consider the inductance of the coils ($L_{k}$). The inductance effect attenuates the time-varying current of the coils. Thus, a small inductance is advantageous for generating a strong time-varying magnetic field. Because the inductance is proportional to $N_{k}$, we can obtain a small inductance by reducing $N_{k}$. However, this also reduces $R_{k}$ in Equation (6) by increasing $A_{k}$ and decreasing $l_{k}$ to keep $T_{k}$ constant. Thus, we can minimize the inductance by minimizing the resistance. However, we should consider the range of $R_{k}$ to utilize the maximum power of the power supply unit. Figure 4 shows the output range of the power supply unit (3001iX by California Instruments); the resistance must be in the range of 8.62 Ω ≤ $R_{k}$ ≤ 19.05 Ω to utilize the maximum power. Then, the available minimum resistance becomes 8.62 Ω to minimize inductance of each coil, and we obtain the following resistance constraint:(16)$$R_{k}=8.62\;\Omega .$$

We also constrained $t_{k}$ because a wire that is too thick is difficult to wind. We experientially introduce the following constraint:(17)$$t_{k}<2\;\mathrm{mm}.$$

### 3.3. Algorithm to Optimize the MNS

We developed an optimization algorithm using Equations (10) and (11) and the given constraints, shown in Figure 5. We set the range of the two design variables as $0.001\leq t_{k}\leq 2$ and $1\leq N_{k}<1420$. The range of $t_{k}$ was determined using Equation (17), and its minimum incremental value was 0.001 mm. The maximum number of turns of wires between the pipes was 1420 when $t_{k}$ had a minimum value of 0.001 mm. Thus, the maximum limit of $N_{k}$ was 1420. Using this algorithm, we obtained the optimized MNS as shown in Table 1. Because the mathematic functions in Equations (10) and (11) were utilized for the algorithm, the calculation time for each step was very short. Thus, the results of the optimization could be obtained in a few seconds.

## 4. Experiments

### 4.1. Construction of the MNS

The optimized MNS was constructed, as shown in Figure 6, and the major variables were measured, as shown in Table 2. Each coil was connected to a power supply unit (3001iX, California Instruments). A magnetic robot inside the MNS was tracked using a real-time camera and was controlled by a joystick controller. The two pipes for the coils were made of fiber-reinforced plastic that could withstand the weight and heat of the coils. Plastic is a nonmetallic material; therefore, the pipes have no iron loss that attenuates the magnetic field. In contrast, the structure used to support the pipes was made of metallic aluminum. Although aluminum can cause iron loss, it does not affect the magnetic field inside the MNS because the structure is located outside the coils.

The constructed MNS had considerable geometrical errors because there was no available winding machine for the coils, owing to their unusual size and shape. In particular, the circular HC and MC were wound with the larger radius ($r_{k}$) than the designed value. This is because they were attached to the outside of the circular pipe. If the radius ($r_{k}$) is smaller than the designed value, they cannot be assembled with the pipe. Thus, a margin was added to manufacture the HC and MC. As a result, these errors can be reflected by the correction factors, which are the ratios of the measured and designed values. Table 3 lists the correction factors; the correct $B_{k}$ or $G_{k}$ can be obtained by multiplying the correction factor and each designed $B_{k}$ or $G_{k}$, respectively.

We also measured the magnetic field near the center of the MNS to verify the spatial homogeneity of the magnetic field. During the experiment, a current of 1 A was applied to each coil, and the magnetic field was measured using a Gauss meter (Model 8030 by F. W. Bell). We calculated the magnetic field of each coil considering the correction factor, and Figure 7 shows the comparison between the calculated and measured magnetic field. We confirmed that the magnetic fields of the HC and MC matched well with the calculated values. However, the saddle coils had relatively large error at the outside of the center because they could not be ideally constructed due to their complex geometry. In particular, the maximum error of 2.9% was measured at the USCy which had the maximum thickness of wire.

### 4.2. Heating Effect

The electrical insulation system for the wires was divided into different classes by temperature; we used an F-class wire for each coil. The F-class wire has an allowable temperature of 155 °C; this must be a temperature limit for safe use. The temperature rise due to the current of the coils was measured using an infrared thermometer, as shown in Figure 8, while the maximum output power was utilized. To obtain an average value, the temperature was measured at six points on each coil. Only USCz2 demonstrated an unlimited operating time because its temperature converged at 96 °C, while the temperatures of the other coils rose over 155 °C. In the figure, the USCz1 had the minimum operating time limit of 16 min, which can be considered the operating time limit of the MNS. However, when a magnetic robot is actuated, each coil operates discontinuously below the maximum output power. Thus, the practical operating time limit of the MNS would be longer than 16 min. For example, if we generate a 2D rotating magnetic field of 15 mT in the *xy*-plane, the operating time limit would be much longer than 40 min because we discontinuously utilize the HC and USCy below the maximum output power.

### 4.3. Inductance Effect

Each coil was designed to have a minimum inductance within the maximum output range of the power supply unit. The current drop due to the inductance effect can be expressed as follows [21]:(18)$$I_{k}=\frac{V_{k}}{\sqrt{R_{k}^{2}+{\left(2\pi L_{k}f_{k}\right)}^{2}}}\;,$$
where $V_{k}$ and $f_{k}$ are the input voltage and frequency of the power supply unit for the *k*-th coil, respectively. Because the magnetic field is proportional to the current in Equation (3), the calculated maximum magnetic field and field gradient in Table 1 cannot be obtained with this current drop. Figure 9 shows the measured maximum magnetic field and field gradient with a variation in the frequency. In particular, the amplitude of each maximum magnetic field decreased to 50% or less at 20 Hz. Thus, this inductance effect should be considered if we generate a time-varying magnetic field.

### 4.4. Comparison to the Conventional System

To verify the optimized MNS, we compared it with the conventional MNS in [21,22], as shown in Table 4 and Table 5. We selected this conventional MNS because it had an identical maximum output power and similar outer and inner diameters to the optimized MNS. As shown in the tables, each coil of the optimized MNS generated a larger magnetic field than that of the conventional MNS. In particular, the maximum 3D rotating magnetic field increased by 42%. However, the magnetic field gradient of the optimized MC decreased. Instead, it had half the inductance as before. This is because the conventional MC was designed to have a relatively large number of turns. Although several inductances of the optimized MNS increased, the maximum inductance of the MNS was reduced by 53% from 859.9 mH to 405.1 mH, which allowed the optimized MNS to generate a larger time-varying magnetic field and field gradient. We also observed an increase in the operating time limit owing to the enlarged cross-sectional area of each coil. If we consider the extreme condition in which the five coils are simultaneously utilized with the maximum output power, the optimized MNS can be operated for seven times longer than the conventional MNS.

### 4.5. Performance Test Using a Rotating Magnetic Field

The maximum 3D rotating magnetic field of the optimized MNS was 42% larger than that of the conventional MNS in the static state. This improvement of static magnetic field can enhance the quasi-static motions such as steering motion of a magnetic catheter and sampling motion of a magnetic capsule [2,16,31]. In addition, the optimized MNS can generate a stronger time-varying magnetic field with a smaller inductance effect, because the inductance was improved by 53%. This improvement can also enhance the dynamic motion of magnetic robots. As one example, we compared the unclogging ability of a helical robot in each MNS, as shown in Figure 10. First, the step-out frequencies of the helical robot were measured using the maximum 3D rotating magnetic fields of the two MNSs. The measured step-out frequency of the optimized MNS was 18 Hz, which is 38% higher than the 13 Hz of the conventional MNS. These frequency differences may affect the drilling ability of the robots. As shown in Figure 10b, the helical robot was actuated in front of the clogged area using agar. Considering the frictional energy consumption during the unclogging motion, slightly lower frequencies (17 Hz and 12 Hz) than the step-out frequencies (18 Hz and 13 Hz) were used for each experiment. As a result, the unclogging time improved by 54% from 96 s to 44 s. Because the propulsive force of a helical robot is proportional to the square of the robot dimension [32], larger robots would exhibit a better improvement. In particular, for the magnetic robots actuated in a fluidic flow such as blood flow in a vessel, their drag forces are proportional to the square of the fluidic velocity [24]. Thus, this improvement of the 3D rotating magnetic field would significantly help the magnetic robots to accomplish their mission.

## 5. Conclusions

In this paper, we proposed an electrical optimization method based on a novel arrangement of the MNS with saddle coils. The MNS was optimized to generate a greater magnetic field and field gradient than a conventional MNS with the same output power and similar size. As a result, each coil could generate an average of 22% stronger magnetic field or field gradient, and the maximum 3D rotating magnetic field was improved by 42%. The proposed optimization method also minimized the current density and inductance of the coils so that the MNS could generate a stronger time-varying magnetic field, with at least sevenfold longer operating time. The optimized MNS could effectively enhance the magnetic robots. As one example, we demonstrated that the unclogging performance of a helical robot was improved by 54%.

Through this study, we verified that, even if the coil is geometrically optimized, the performance of the coil can be limited in the case that the electrical characteristics are not properly designed. In contrast to coreless MNS, many researchers have studied the optimization method of MNSs with a core [15,16,17], because the performance of the MNSs with a core can significantly vary depending on various factors of the core such as the ratio between the core and coil, material property of the core, and shape of the core tip. However, they did not focus on the electrical characteristics of the MNSs. We believe that this electrical optimization can be applied to MNSs both with and without cores.

The proposed optimization method increases the cross-sectional area of the coils to suppress the temperature rise of the coils by reducing the current density. However, this may increase the material cost because copper coil is quite expensive. Thus, an excessively large cross-sectional area of the coil is not recommended in terms of cost. Although the thickness of the wire was limited, it was still too thick to manufacture complex coils without manufacturing errors; therefore, the thickness of wires should be carefully determined considering the shapes and volumes of the coils.

## Figures and Tables

**Figure 1 sensors-22-05603-f001:**
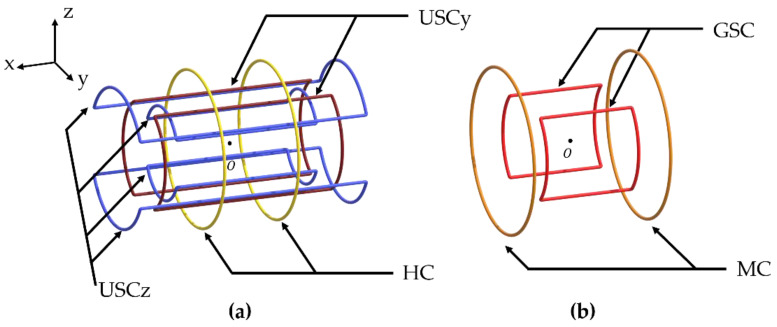
The proposed novel arrangement of five pairs of coils to enable the MNS to generate the maximum 3D rotating magnetic field: (**a**) three pairs of coils (HC, USCy, and USCz) generate the uniform magnetic field; (**b**) two pairs of coils (MC and GSC) generate the uniform magnetic field gradient.

**Figure 2 sensors-22-05603-f002:**
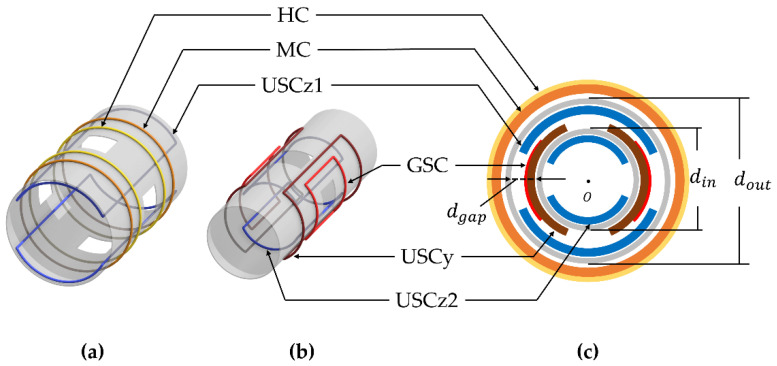
The proposed arrangement of the pipes (frame) and five pairs of coils: (**a**) HC and MC are installed outside the outer pipe, and USCz1 is installed inside the outer pipe; (**b**) USCy and GSC are installed outside the inner pipe, and USCz2 is installed inside the inner pipe; (**c**) the five pairs of coils integrated with the two pipes.

**Figure 3 sensors-22-05603-f003:**
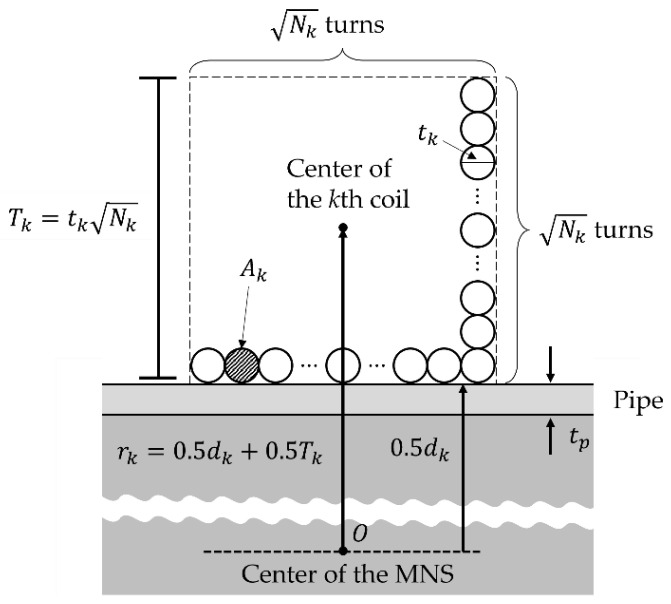
The design variables and parameters of the *k*-th coil. This cross-section shows the case in which the coil is attached to the outside of the pipe.

**Figure 4 sensors-22-05603-f004:**
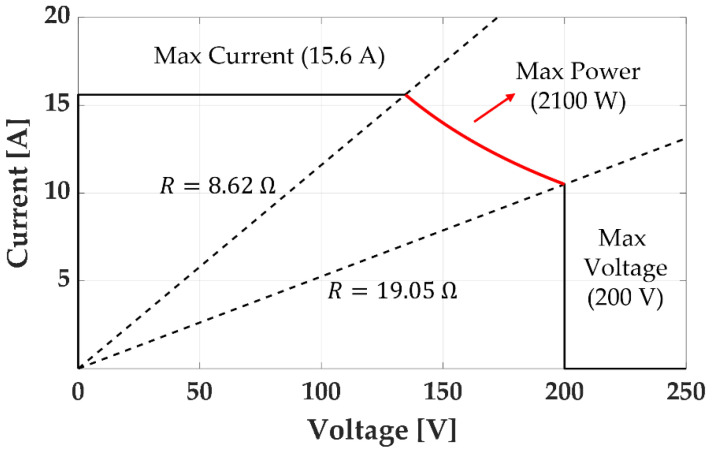
Output range of the power supply unit (3001iX by California Instruments).

**Figure 5 sensors-22-05603-f005:**
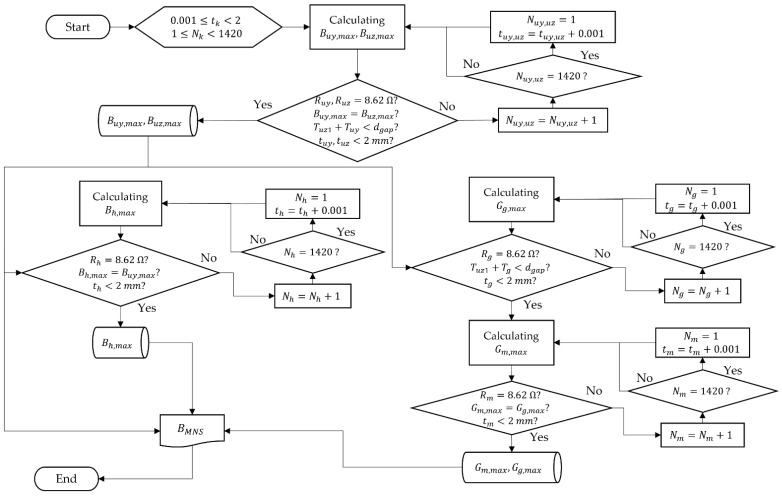
Optimization algorithm for the MNS with the given constraints.

**Figure 6 sensors-22-05603-f006:**
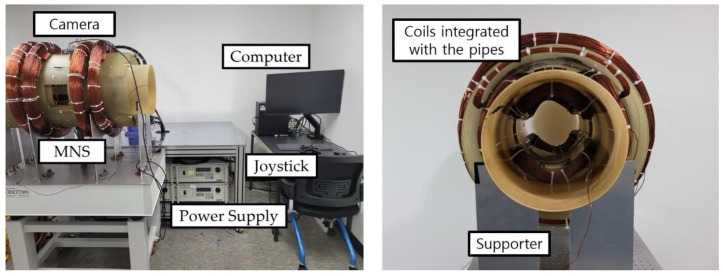
The constructed MNS and experimental setup to actuate a magnetic robot.

**Figure 7 sensors-22-05603-f007:**
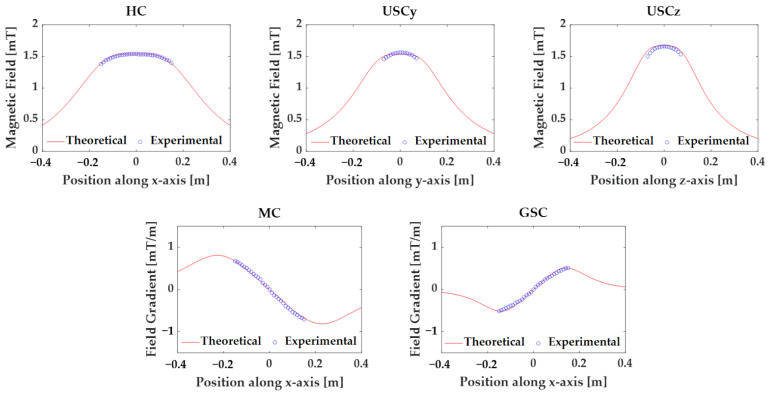
Calculated and measured magnetic field distribution along the axis of each coil when the current of 1 A was applied.

**Figure 8 sensors-22-05603-f008:**
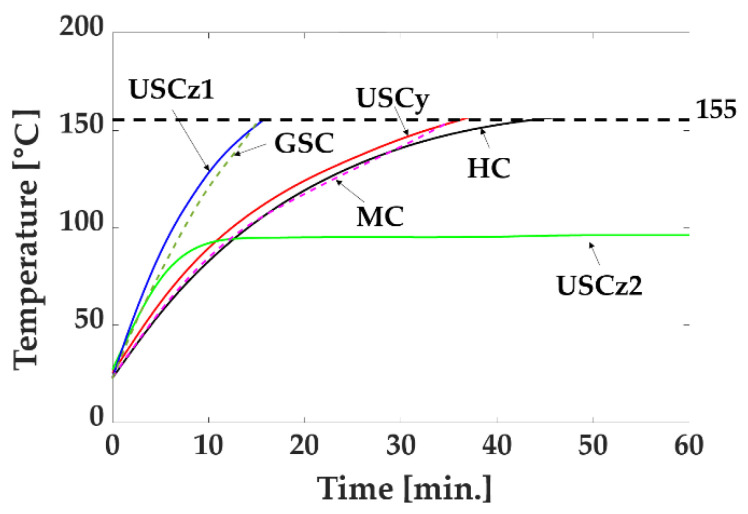
Temperature rise of coils for elapsed time.

**Figure 9 sensors-22-05603-f009:**
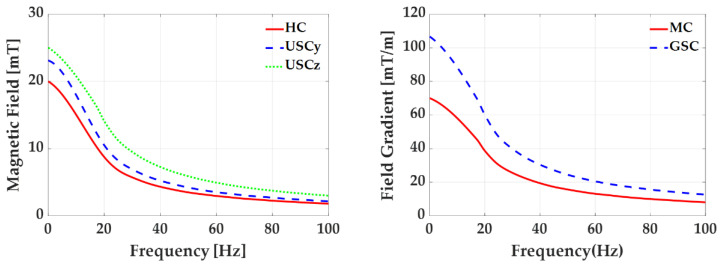
Calculated and measured maximum magnetic field or field gradient of each coil with a variation of frequency.

**Figure 10 sensors-22-05603-f010:**
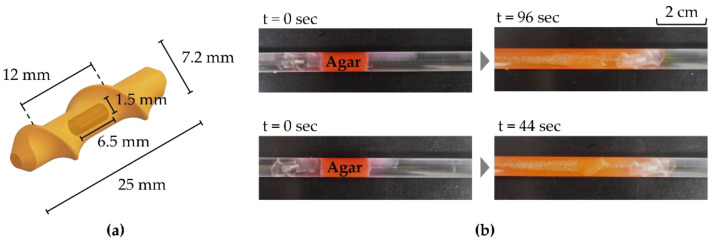
Performance test using a rotating magnetic field: (**a**) helical robot made of a diametrically magnetized cylindrical magnet (N52 grade) for the demonstration; (**b**) unclogging motion of the helical robot inside the optimized and conventional MNS. The optimized MNS and conventional MNS generated a 3D rotating magnetic field at 17 Hz and 12 Hz, respectively.

**Table 1 sensors-22-05603-t001:** Major variables of the optimized MNS.

Variables	HC	USCy	USCz1	USCz2	MC	GSC
$\mathrm{Radius}\;\mathrm{of}\;\mathrm{the}\;\mathrm{coil}\;(r_{k}$) (mm)	249.8	173.1	220.0	137.0	248.2	173.2
$\mathrm{Turns}\;\mathrm{of}\;\mathrm{the}\;\mathrm{wire}\;(N_{k}$) (turns)	448	370	40	269	413	479
$\mathrm{Thickness}\;\mathrm{of}\;\mathrm{the}\;\mathrm{wire}\;(t_{k}$) (mm)	1.86	1.89	1.59		1.79	1.65
$\mathrm{Resistance}\;\mathrm{of}\;\mathrm{the}\;\mathrm{coil}\;(R_{k}$) (Ω)	8.62	8.62	8.64		8.62	8.63
$\mathrm{Max}.\;\mathrm{magnetic}\;\mathrm{field}\;(B_{k,max}$$)\;(\mathrm{mT})\;\mathrm{or}\;\mathrm{field}\;\mathrm{gradient}\;(G_{k,max}$) (mT/m)	25.16	25.15	25.26		84.26	102.99

**Table 2 sensors-22-05603-t002:** Measured major values of the MNS.

Variables	HC	USCy	USCz1	USCz2	MC	GSC
$\mathrm{Radius}\;\mathrm{of}\;\mathrm{the}\;\mathrm{coil}\;(r_{k}$) (mm)	262.3	175.0	220.0	137.1	260.7	170.0
$\mathrm{Resistance}\;\mathrm{of}\;\mathrm{the}\;\mathrm{coil}\;(R_{k}$) (Ω)	10.2	9.8	9.0		10.1	9.0
$\mathrm{Inductance}\;\mathrm{of}\;\mathrm{the}\;\mathrm{coil}\;(L_{k}$) (mH)	405.1	364.6	170.2		303.0	248.5
$\mathrm{Max}.\;\mathrm{magnetic}\;\mathrm{field}\;(B_{k,max}$$)\;(\mathrm{mT})\;\mathrm{or}\;\mathrm{field}\;\mathrm{gradient}\;(G_{k,max}$) (mT/m)	20.00	23.11	25.02		70.04	106.67

**Table 3 sensors-22-05603-t003:** Correction coefficients of the MNS.

	HC	USCy	USCz	MC	GSC
Correction coefficients	0.9045	0.9540	1.0313	0.8623	1.0739

**Table 4 sensors-22-05603-t004:** Comparison of major values between the optimized and conventional MNS.

Variables		Conventional MNS	Optimized MNS	Differences [%]
Max magnetic field $\left(B_{k,max}\right)$ (mT)	HC	14.18	20.00	41
	USCy	21.69	23.11	7
	USCz	14.04	25.02	78
Max magnetic field gradient $\left(G_{k,max}\right)$ (mT/m)	MC	121.3	70.04	−42
	GSC	83.70	106.67	27
Inductance $\left(L_{k}\right)$ (mH)	HC	344.5	405.1	18
	USCy	201.3	364.1	81
	USCz	394.3	170.2	−57
	MC	859.9	303.0	−65
	GSC	84.60	248.5	194
Operating time limit with max. power (min)	HC	7	60	757
	USCy	15	40	167
	USCz	10	16	60
	MC	14	35	150
	GSC	2	17	750
Diameter of an MNS (mm)	Outer	470	526	12
	Inner	235	240	2
Max power for each coil (W)		2100	2100	-

**Table 5 sensors-22-05603-t005:** Comparison of practical values between the optimized and conventional MNS.

Variables	Conventional MNS	Optimized MNS	Differences (%)
Max. 3D rotating magnetic field (mT)	14.04	20.00	42
Min. magnetic field gradient (mT/m)	83.70	70.04	−16
Max. inductance (mH)	859.9	405.1	−53
Min. operating time limit of the coils with max. power (min.)	2	16	700

## Data Availability

Not applicable.

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
