# Peer review of "Electrical Optimization Method Based on a Novel Arrangement of the Magnetic Navigation System with Gradient and Uniform Saddle Coils"

_sensors, 2022, doi:10.3390/s22155603_

Round 1

Reviewer 1 Report

The present manuscript presents a new method for optimum magnetic navigation of magnetic robots. The manuscript is well-written and the conclusions are supported by the results. On the other hand, there are some points in the manuscript that needs to be improved before the publishing. The following comments should be taken into account.

1. The originality of the paper needs to be stated clearly. It is of importance to have sufficient results to justify the novelty of a high-quality journal paper. The Introduction should make a compelling case for why the study is useful along with a clear statement of its novelty or originality by providing relevant information and providing answers to basic questions such as: What is already known in the open literature? What is missing (i.e., research gaps)? What needs to be done, why and how? Clear statements of the novelty of the work should also appear briefly in the Abstract and Conclusions sections.

2. Authors should add more explanation about this investigation contribution in the introduction section.  

3. The results are not discussed in the manuscript. Authors should discuss the results and how they can be interpreted in perspective of previous studies and of the working hypotheses. The findings and their implications should be discussed in the broadest context possible and limitations of the work highlighted. 

4. How fast is the method? How long does it take in order to calculate the appropriate values of the magnetic field? Relative comments should be included in the manuscript.

5. How the velocity of the fluid affects the results of the method? Relative comments should be included in the manuscript.

6. How the size of the robots affects the results of the methodology? Relative comments should be included in the manuscript.

7. How the method manages two robots? Especially when these robots are not in the same location.

8. How the navigation of the robots affected by an increase on the steady magnetic field ?

9. The introduction section need to be improved in order to cover other methods for the magnetic navigation. The following research articles may be included in the manuscript:

a)      A computational tool for the estimation of the optimum gradient magnetic field for the magnetic driving of the spherical particles in the process of cleaning water, Desalination and Water Treatment, 99, 27 – 33.

b) Simultaneous Magnetic Particle Imaging and Navigation of large superparamagnetic nanoparticles in bifurcation flow experiments, Journal of Magnetism and Magnetic Materials, 498, 15, 166206.

c)   Development of a real time imaging-based guidance system of magnetic nanoparticles for targeted drug delivery, Journal of Magnetism and Magnetic Materials, 427, 345-351.

d)   Swarm of magnetic nanoparticles steering in multi-bifurcation vessels under fluid flow, Journal of Micro-Bio Robotics, 16, 137–145.

         e)   Magnetic Resonance Navigation for Targeted Embolization in a Two-Level Bifurcation Phantom, Annals of Biomedical Engineering, 47, 12.

Reviewer 2 Report

In this paper, the authors propose a methodology for optimizing a magnetic navigation system for controlling micro robots. The structure of the system repeats that described in a number of previous studies. Optimization is performed according to the parameters of the number of turns in coils and the thickness of the wire. There are the following remarks about the work:

1. From the above description of the optimization algorithm, one gets the impression that the optimization is done by maximizing the magnitude of the field created by the various coils in the system. It is not clear how the inductance is optimized. Since the system is designed to create rotating fields among other things, the inductance of the coils must be minimized to allow the creation of relatively high frequency magnetic fields. The authors indicate that they were able to increase the frequencies of the magnetic field, but exactly how this was achieved is not clear. The authors should describe their optimization algorithm in more detail, taking into account not only the values of the generated field, but also the inductance of the coils.

2. Formulas (10), (11) do not take into account changes of coils resistance with time under the action of current. They are suitable for calculating the fields only in the initial moment, this should be explicitly pointed out in the article.

3. Did the degree of spatial homogeneity of the magnetic field and the magnetic gradient change as a result of the optimization performed? There is no information in the paper about measurements of magnetic field in the system.

4. In the paper temperature is measured in degrees (line 225 and Fig. 7). A specific physical quantity should be used, for example Celsius degrees (°C).

Overall, the paper contains some new material and can be published after the mentioned remarks are corrected.

Round 2

Reviewer 1 Report

The manuscript can now be accepted for publication

Reviewer 2 Report

The authors have taken into account the comments and, as presented, the work can be recommended for publication.